# Tetraspanin CD9 Expression Predicts Sentinel Node Status in Patients with Cutaneous Melanoma

**DOI:** 10.3390/ijms23094775

**Published:** 2022-04-26

**Authors:** Guendalina Lucarini, Elisa Molinelli, Caterina Licini, Giulio Rizzetto, Giulia Radi, Gaia Goteri, Monica Mattioli-Belmonte, Annamaria Offidani, Oriana Simonetti

**Affiliations:** 1Dipartimento di Scienze Cliniche e Molecolari-Istologia, Università Politecnica delle Marche, 60126 Ancona, Italy; guendalina.lucarini@univpm.it (G.L.); c.licini@univpm.it (C.L.); m.mattioli@univpm.it (M.M.-B.); 2Clinica Dermatologica, Dipartimento di Scienze Cliniche e Molecolari, Università Politecnica delle Marche, 60126 Ancona, Italy; molinelli.elisa@gmail.com (E.M.); g.rizzetto@univpm.it (G.R.); g.radi@univpm.it (G.R.); o.simonetti@univpm.it (O.S.); 3Anatomia Patologica, Dipartimento di Scienze Biomediche e Sanità Pubblica, Ospedali Riuniti, Università Politecnica delle Marche, 60126 Ancona, Italy; g.goteri@univpm.it

**Keywords:** tetraspanin CD9, melanocytic nevi, cutaneous melanoma, lymph node metastasis, transendothelial invasion, prognostic marker

## Abstract

The tetraspanin CD9 is considered a metastasis suppressor in many cancers, however its role is highly debated. Currently, little is known about CD9 prognostic value in cutaneous melanoma. Our aim was to analyse CD9 expression in melanocytic nevi and primary cutaneous melanomas through immunohistochemistry and immunofluorescence approaches to determine its correlation with invasiveness and metastatic potential. CD9 displayed homogeneous staining in all melanocytic nevi. In contrast, it showed a complete loss of reactivity in all thin melanomas. Interestingly, CD9 was re-expressed in 46% of intermediate and thick melanomas in small tumor clusters predominantly located at sites of invasion near or inside the blood or lymphatic vessels. The most notable finding is that all CD9 stained melanomas presented sentinel node positivity. Additionally, a direct association between CD9 expression and presence of distant metastasis was reported. Finally, we confirm that CD9 expression is consistent with an early protective role against tumorigenesis, however, our data endorse in melanoma a specific function of CD9 in vascular dissemination during late tumor progression. The presence of CD9 hotspots could be essential for melanoma cell invasion in lymphatic and endothelial vessels. CD9 could be a valid prognostic factor for lymph node metastasis risk.

## 1. Introduction

Malignant melanoma, originated from genetically modified or activated melanocytes, can be considered a result of complex interactions between environmental, constitutional, and genetic factors. Melanoma is characterized by clinically distinguishable subtypes: cutaneous, mucosal, uveal, and unknown primary melanomas [1,2,3,4].

Once a rare tumor, cutaneous melanoma is considered the third most common cancer. Despite the increasing recognition of melanoma in situ, the incidence of the invasive melanoma was not decreased [5]. In Europe, the melanoma incidence rate is <10–25 new cases per 100,000 inhabitants [6], with the highest percentage of fatal cases seen in men over the age of 65 years old and with thick lesions (>1 mm) [7]. Thus, early diagnosis and treatment innovations, encompassing targeted therapies and immune checkpoint-inhibitors, promote better outcomes [8], taking into account that responses to the treatments depend on the efficiency of the host immune system, including the innate immunity [9,10,11]. Recent research has highlighted the importance of the melanoma microenvironment, capable of influencing melanoma progression and its response to anticancer therapies [12,13]. 

Even if a real decline in mortality rates has been recorded in the past few years [14], the rapid rise in cutaneous melanoma remains one of the most aggressive tumors in the world. Therefore, exploring the molecular mechanisms subjacent melanoma development could be helpful to identify biomarkers for reliable prediction of survival and recurrence.

Tetraspanins, a family of transmembrane proteins, have been confirmed to play an important role in carcinogenesis and malignant progression [15]. The tetraspanin CD9 is regarded as a metastasis suppressor in some tumors where low levels of this protein correlate with advanced disease and poor prognosis. An early study by Ikeyama et al. demonstrated that CD9 overexpression decreased their motility in a variety of cancer cells [16]. In many tumors such as breast, lung, and colon, lower CD9 expression worsens the prognosis. Nevertheless, some studies suggested opposite results [17]. In acute lymphoblastic leukemia, CD9 expression was associated with a poor prognosis [18] as well as its overexpression in an aggressive tumor like esophageal squamous cell correlated with tumor clinical staging and lymph node involvement [19]. Tasdemir et al. found that gastric carcinoma behaved more aggressively and is inclined to lymphatic and vascular invasion, lymph node metastasis, peritoneal dissemination, and advanced stage, when CD9 is more expressed [20]. Moreover, Miki et al. showed increased migration and invasion abilities in CD9-positive exosomes from cancer-associated fibroblasts of scirrhous-type gastric cancer cells, and the prognosis of patients with positive CD9 expression was worse than CD9-negative patients. [21]. Recently, CD9 staining was also found increased and strongly associated with worse prognosis in pancreatic ductal adenocarcinoma [22] and in primary ovarian tumors [23]. Therefore, the role of CD9 is highly debated and controversial, since its expression in cancer cells has been reported to exert both pro- and anti-migratory functions, probably due to its modulatory activity toward integrins and other transmembrane proteins [24]. 

Tetraspanins, like CD9, often combine with different molecules they assemble into extracellular vesicles (EVs) called exosomes [25,26], whose levels resulted as increased in cancer patients [27]. In an interesting study, Peinado et al. observed that tumor-derived EVs were shown to regulate systemic metastasis by forming pre-metastatic niches in distant organs [28]. 

However, most of these findings have been obtained using tumor cell lines, whereas little is known about CD9 prognostic value. The aim of our study was to analyze CD9 expression in melanocytic nevi and primary cutaneous melanomas through an immunohistochemistry approach to determine its correlation with melanoma invasiveness and metastatic potential.

## 2. Results

### 2.1. Patient Characteristics

All available details of clinical and histopathological features of the melanoma patients are summarized in Table 1.

In total, 20 melanocytic nevi and 120 primary cutaneous melanomas (24 thin, 56 intermediate, and 40 thick melanomas) were included in our study. Median age of the patients was 68 years (range: 30–92 years) with 75 (53.6%) male patients. Median Breslow thickness of the primary melanoma was 2.9 mm (range 0.4–10 mm). Of the 120 melanomas 8 had II, 16 had III, 64 had IV, and 32 had V Clark level.

Tumor ulceration was identified in 12 intermediate (21.4%) and in 22 thick melanomas (55%). There was a significant relationship between Breslow thickness and ulceration presence (*p* < 0.01, Chi square test).

In the 96 patients with intermediate and thick melanoma, sentinel lymph node biopsy (SNB) was performed. Among these patients 44 (45.8%) were positive by hematoxylin-eosin or immunohistochemistry for micro-metastases, or both. Table 1 shows the distribution of positive sentinel lymph nodes (N_1_) according to melanoma thickness. Sixteen (29%) of 56 patients with intermediate melanomas and 28 (70%) of 40 patients with thicker lesions (>4 mm) had a positive sentinel node. Chi square test was utilized to determine the relationship between increment of melanoma thickness and sentinel lymph node status (*p* < 0.001). A logistic regression model was used for describing and analyzing risk factors for SNB positive status in the patients with intermediate and thick melanomas. The results in Table 2 showed that Breslow thickness >4 mm, tumor ulceration, and CD9 positivity were associated with significantly increased risk for SNB positivity. Among these factors, tumor ulceration and CD9 expression were the strongest predictors of positive SNB results (*p* < 0.0001).

In the 120 patients with invasive melanoma, 45 (37.5%) developed distant metastasis after a median of 76 months (95% CI 59–92 months). All patients affected by thin melanoma displayed no evidence of distant metastasis and none of them died for melanoma. Of the 56 patients with intermediate melanomas, 20 (35%) developed distant metastasis after a median time of 105 months (95% CI 87–124 months), compared to 57 months (95% CI 36–78 months) for the 25 of the 40 patients (62%) with thick melanomas (log-rank *p* = 0.006; Figure 1a). Metastasis localizations were in the lung (*n* = 23), liver (*n* = 10), brain (*n* = 8), and skin (*n* = 4). 

Sixteen patients (28%) affected by intermediate melanoma died from melanoma disease (MSS) after a median time of 116 months (95% CI 98–134 months), while 13 patients (32%) with thick lesions died after a median time of 76 months (95% CI 64–87 months). Kaplan–Meier curves revealed that Breslow thickness was significantly correlated with MSS (log rank *p* = 0.02; Figure 1b). 

In Figure 1c, Kaplan–Meier curves showed that patients with intermediate/thick melanoma and positive SNB status (SNB+) died for MSS after a significantly decreased median time compared to the patients affected by intermediate/thick melanoma and negative SNB (SNB-) (log rank: *p* = 0.001; intermediate SNB- median survival time 148 months, 95% CI 137–159 months; intermediate SNB+ 64 months, 95% CI 43–85 months; thick SNB- 94 months, 95% CI 62–127 months; thick SNB+ 64 months, 95% CI 46–81 months). 

The patients with melanoma presenting ulceration were 34 and had median MMS time of 87 months (95% CI 69–105 months), significantly shorter than MSS time of the patients without ulceration (138 months, 95% CI 125–152 months, log-rank *p* = 0.014, Figure 1d).

At multivariate analysis with Cox proportional hazard model, tumor ulceration, CD9 expression, and SNB status resulted in independent prognostic factors associated with MSS (Table 3).

### 2.2. Immunohistochemical and Immunofluorescence Evaluation

The cells forming the basal and suprabasal layers of the uninvolved squamous epithelium displayed strong CD9 staining and provided internal positive control for CD9 immunoreactivity that was pronounced in the basal layer at the regions adjacent to stromal tissue (Figure 2a). 

CD9 expression in cells forming the squamous epithelium from which the tumors arose was not always conserved in melanomas (Figure 2b). CD9 was predominantly localized in the cell plasma membrane (as expected for a membrane antigen), but it was also sometimes detected in the cytoplasm (Figure 2c). 

CD9 showed a marked and homogeneous expression in all melanocytic nevi (Figure 2d–f). In contrast, CD9 was not found in almost two-thirds of the melanomas (76/120). Primary melanomas showed two variations of CD9 staining: (a) an absolute loss of reactivity in the total of thin melanomas (24/24) (Figure 2g–i); (b) a reactivity limited to <50% of the total intermediate and thick melanomas, occurring in 44 of the 96 lesions (Table 4). 

In positive melanomas, the staining pattern was somewhat lower and less homogeneous with respect to the nevi, often turning from “membranous” to “cytoplasmatic”. CD9 staining was mainly confined to several small clusters of tumor melanocytic cells that strongly expressed CD9, even in weakly stained sections or with wide negative regions (Figure 2l–o). These tumor sites with marked localized CD9 expression were detected at a density of at least three per sample in the positive intermediate and thick melanomas. Tumor sections with no CD9 positive clusters were classified as negative (Figure 2p,q).

As shown in microphotographs of immunofluorescence and immunohistochemistry, we successfully stained CD9 positive clusters of melanoma cells, obtaining similar staining patterns with both techniques. As expected, CD9 expression was observed predominantly on the cytoplasm of melanoma cells in positive clusters often located in totally negative tumor areas (Figure 3a).

Immunohistochemical double staining with antibodies against CD9 and CD34 or D240 revealed an interesting finding: CD9 positive clusters, in which CD9 appeared to be re-expressed, were predominately located at sites of invasion near (Figure 4a–c,g–i) or inside the blood (CD34+, Figure 4d–f) or lymphatic vessels (D240+, Figure 4l–n). In summary, CD9 was found to be downregulated or absent in the main tumor mass, but it was still expressed immediately adjacent or within vessels.

Double immunostaining (dark staining) also revealed that CD9 and CD34 are found contemporaneously in the endothelium of the vessels (Figure 4d,e), just as CD9 and D240 in lymphatic vessels (Figure 4l,m), in particular, adjacent to CD9 positive tumoral clusters. The Figure 4f,n showed the immunolocalization of CD9 (brown staining) in endothelial and lymphatic vessels, respectively. This result was confirmed by double immunofluorescence reaction: Figure 3b shows that endothelial vessels (CD34+) and lymphatic vessels (D240+) were both positive for CD9. This marker colocalization was represented by an orange staining.

### 2.3. CD9 Staining and Correlation with Clinic-Pathological Features

The results are summarized in Table 4. We detected the expected decrease in expression in melanoma lesions as compared to nevi. Statistical analysis revealed that CD9 staining was directly related to the thickness of the primary melanomas (*p* < 0.0001, Figure 3c); it was absent in all thin melanomas, while 16 cases of 56 intermediate melanomas (29%) and 28 of 40 thick melanomas (70%) showed CD9 positive clusters (*p* < 0.05; Figure 3d). Similarly, CD9 was lost in melanomas with II and III Clark level and re-expressed in 20 of the 64 melanomas with IV level (31%) and in 24 of the 32 with V Clark level (75%). These differences in CD9 expression were statistically significative (*p* < 0.05; Figure 5a) and supported by the existence of a significant directly correlation between CD9 positivity and Clark level (*p* < 0.001; r = 0.753; Figure 5b).

The most interesting finding was that all 44 CD9 stained melanomas (100%) presented sentinel node positivity (CD9+N_1_) and, consequently, none of the CD9 negative melanomas (CD9-N_0_) had sentinel lymph node micrometastases. Therefore, CD9 expression was closely associated with the sentinel lymph node status (*p* < 0.001; Figure 5c). As shown in Table 2, logistic regression analysis revealed that CD9 expression was a strong predictor of SNB positivity (*p* < 0.0001).

Comparing the differential expression patterns of CD9 with distant metastasis status and considering that all patients affected by thin melanomas showed no evidence of disease progression, it was found that 29 of the 45 primary intermediate and thick melanomas with distant metastasis (64%) were stained for CD9 (CD9+M1) while only 15 CD9 positive patients had no metastasis (CD9+M_0_) (*p* < 0.001). On the contrary, most CD9 negative patients (71%) developed no metastasis (CD9-M_0_) (*p* < 0.001; Figure 5d). 

To evaluate whether marker expression correlated with prognosis of melanoma patients, Kaplan–Meier survival curves were considered using melanoma-specific survival (Figure 5e). The analysis displayed that the patients positive to CD9 had a significantly shorter MSS (median 98 months, 95% CI 80–116 months) compared to the negative patients (median 130 months, 95% CI 116–145 months; log rank *p* = 0.018). Cox regression analysis for MSS showed that CD9 expression, along with tumor ulceration and sentinel node status, was an independent predictor of melanoma specific survival in patients with cutaneous melanoma (*p* < 0.001; Table 3).

## 3. Discussion

It is now recognized that prognostic and clinicopathological significance of CD9 expression is controversial and changed with respect to the tumor type. In fact, the tetraspanins have recently gained attention as both suppressors and promoters of metastasis. It can be presumed that variability in the membrane and vesicular elements, associated with single tetraspanins, justifies their conflicting skills to promote and suppress metastasis [26]. In particular, CD9 tends to interact with various integrins and transmembrane proteins [24] and, together with other tetraspanins such as CD63 and CD81, is recognized as an abundant marker on the surface of exosomes [29,30]. An interesting study showed that extracellular vesicles (EVs), in particular exosomes, play an essential role in both primary tumor growth and metastatic progression, since they are critical mediators of intercellular link between tumor cells and stromal cells in processes like vascular leakiness, extracellular remodeling, and regulation of the immune system [31].

In cutaneous melanoma, a pilot study of Si et al. demonstrated an inverse correlation between CD9 expression and the invasive potential of tumor cells [32]. However, it has been demonstrated that B16 mouse melanoma cells overexpressing CD9 showed increased capacity to invade Matrigel, although CD9 positivity in mouse and human melanoma cell lines was found to be reduced with respect to normal melanocytes [33]. Rambow et al. described the presence of CD9 in the MeLiM swine model during spontaneous regression and differentiation of melanomas. In particular, high CD9 expression was observed in sparse highly pigmented cells, while small pigmented cancer cells were weakly or not stained [34].

In human samples, Mischiati et al. reported that this tetraspanin was expressed in 18/18 nevi but was lost in 20/28 primary melanomas (71%, including thin lesions), which showed a complete loss in more than 90% of the cells or a limited reactivity [35]. Similarly, we observed that CD9 expression was present in all nevi but lost in the totality of thin melanomas and in 52 of intermediate and thick melanomas (63%). Interestingly, CD9 expression was restored in 30% and in 70% of the intermediate and thick melanomas, respectively, even if confined to several small clusters of tumor melanocytic cells. We believe that the presence of CD9 expression in melanocytes of nevi is coherent with a protective function of the tetraspanin against tumorigenesis. Fan et al. hypothesized that in the transition phase from melanocytes to melanoma, the suppression of CD9 amounts may be crucial [33]. However, in most advanced-stage melanomas, as we have observed, CD9 re-expression within the tumor microenvironment may lead to enhanced invasion, suggesting that the decreased of CD9 expression occurs in the earliest stage of melanoma while it is re-expressed in the invasive stage [33]. Interestingly, in our study we found that these melanocytic clusters in which the tetraspanin was re-expressed were located at sites of invasion near or even inside the blood or lymphatic vessels positive for CD9. Similar results were obtained by Sauer et al. in cervical cancer. The authors found that the tumor sites with high localized CD9 positivity showed cones expanding into lymphatic or blood vessels and suggested a tetraspanin functional role in transendothelial migration as a critical step in lymph node metastases development [36]. Consistent with these findings, we found, for the first time in melanoma, that all the specimens presenting lymph node metastasis were CD9 positive. The majority of these samples showed distinct regions of strong immunoreactivity at sites of vessel penetration that were mostly CD9 stained. Erovic et al., after hall, found that this tetraspanin was a valid marker for lymphatic endothelial cells and able to promote transmigration of tumor cells through the adherence to lymphatic vessels [37]. This is also supported by our other important finding, namely that CD9 was associated with significantly increased risk for SNB positivity, as well as tumor ulceration and Breslow thickness. 

It is well known that locoregional lymph nodes metastases set up to a higher risk of progression into distant metastases and recurrence following surgery [38]. Another intriguingly result of our study was a direct correlation between CD9 expression and presence of distant metastasis. This data is consistent with previous studies of Garcia-Lopez et al., showing that melanocyte motility is enhanced by CD9 and suggesting that its overexpression may partly cause the invasion activity of melanoma cells across the Matrigel [39]. In fact, a critical step in the lymphatic or distant metastasis is the invasion of lymphatic or blood vessels by cancer cells. Different types of cell surface glycoproteins, which impact a wide variety of cellular processes, are essential for adhesion, motility, and the ability of tumor cells to invade surrounding tissue [40]. Among these, tetraspanins have a key role as transmembrane adapter proteins forming functionally complexes with adhesion molecules. As already mentioned, CD9 was detected as the most prominent marker of extracellular membrane vesicles, nanometer-sized entities that are released into surrounding body fluids and affect intercellular communication under physiological and pathological conditions [41]. Leary et al. found that EVs derived from melanoma cells are rapidly transported by lymphatic vessels to draining lymph nodes, where they selectively interact with lymphatic endothelial cells [42]. Therefore, in vitro studies on transendothelial migration of melanoma cells highlight the significant role of CD9 in tumor–endothelial/lymphatic cell interaction and vascular dissemination of tumor cells. Ito et al. showed that heterologous gap junctions between endothelial and melanoma cells can help to metastasis formation in vivo [43]. Interestingly, it has been shown that CD9 localized along lateral junctions of endothelial (CD34 positive) and epithelial cells [25]; more specifically, Longo et al., similarly to our findings, described that endothelial cells had an active redistribution of CD9 to the points of melanoma cell insertion, and CD9 expression was mostly concentrated at tumor cells-endothelial cells contact areas, suggesting a stated role of CD9 in the extravasation phase of cancer cell invasion. Moreover, it has been showed that melanoma cells’ transendothelial migration was inhibited by anti-CD9 monoclonal antibodies [44]. In lymphatic endothelial cells, CD9 was also seen to regulate molecular organization of integrins, supporting several functions necessary for lymphatic vessel formation [45]. Our results confirmed that tumor cells CD9 positive may establish intimate contact with endothelial cells. This finding highlights an important role for endothelial/lymphatic CD9 in active recognition required for melanoma cells during insertion. However, the CD9 contribution to the in vivo transcellular migration deserves further investigation.

The presence of tumor cells within patient lymph nodes is an important indicator of poor prognosis and progression. Recent research has showed that tumor cells navigating the lymphoid system acquire advantages in survival, suggesting that metastases in lymph node may be the starting point of distant metastasis. Recently, Ubellacker et al. reported that melanoma cells in lymph incorporate oleic and other antioxidants acid, protecting them from ferroptosis (an iron dependent programmed cell death) and increasing their capacity to survive for subsequent migration through the blood [46]. 

In conclusion, in line with previous reports [47,48], our results confirmed that Breslow thickness, tumor ulceration, and SNB status were significantly associated with poor prognosis for patients with cutaneous melanoma. The direct correlation between CD9 expression and distant metastasis observed in our study assesses a potential clinical value of this tetraspanin in melanoma prognosis. We confirmed that CD9 expression is consistent with an early protective role of the tetraspanin against tumorigenesis, however, our data endorse in melanoma a specific function of CD9 in vascular dissemination during late tumor progression, supported by the existence of CD9 re-expressing cell clusters that mediate transendothelial invasion of the tumor cells. 

Our most important finding shows that the presence of CD9 hotspots is essential for melanoma cell invasion in lymphatic and endothelial vessels. 

Melanoma is an aggressive tumor, in particular intermediate and thick melanomas that easily metastasize. It is well known that positive sentinel node biopsy identifies patients demanding adjuvant therapy. Breslow thickness is usually used to identify patients for SNB. Several clinical pathological characteristics, including Clark level, ulceration, mitotic count, vessel invasion, tumor site, and age, have been associated with SNB positivity [49,50], albeit with conflicting results. Therefore, molecular biomarkers could be an effort to improve melanoma risk stratification and to avoid unnecessary SNB procedures. Recently, several authors suggested the predictive ability of a model that included molecular variables in combination with the clinicopathologic variables (Breslow depth and tumor ulceration) [51,52]. The Clinicopathological and Gene Expression Profile (CP-GEP) model was recently developed to accurately identify patients with primary melanoma at low risk for nodal metastasis [53,54]. Our results showed that CD9 expression exhibited a significant predictive effect on SNB status, suggesting that it could serve as a useful addition to a marker panel for selecting melanoma patients, in particular with intermediate thickness melanoma for SNB.

## 4. Materials and Methods

### 4.1. Sample Collection

All paraffin-embedded tissue samples from benign and malignant melanocytic lesions, surgically removed between 2005 and 2015, were identified retrospectively from the Archives of the Institute of Pathological Anatomy of the Marche Polytechnic University. Diagnoses were made on hematoxylin and eosin-stained sections and based on the usual criteria [55]. Histopathological diagnoses of the included cases comprised 20 ordinary, benign, melanocytic nevi and 120 primary cutaneous melanomas. The inclusion criteria for melanoma patients were as follows: only one primary malignancy, complete follow up, clear causes of death. Exclusion criteria were: patients with more than one primary cancer, patients younger than 18 years old of age, deaths reported within the first month of diagnosis. In order to guarantee a valid sample number, melanomas were selected sequentially until 120 according to Breslow thickness, lymph node, and distant metastasis status. Clinical data of the patients were retrieved from medical records. Twenty-four cases with Breslow thickness ≤ 1.0 mm were assembled in the category of thin melanomas, 56 cases with Breslow thickness 1.1–4.0 mm were considered intermediate melanomas and 40 cases > 4.0 mm were considered thick melanomas. None of the patients had been treated with adjuvant, immune, or targeted therapy. Ninety-six patients (melanoma Breslow > 1.0 mm) underwent sentinel lymph node biopsy (SNB). Lymph node specimens were stained with routine hematoxylin-eosin and assessed by our institution’s dermatopathologists for presence of micrometastasis. The patient was considered to have a positive sentinel node if one or more lymph nodes stained positive for melanoma cells by hematoxylin-eosin staining. Negative nodes were screened for micrometastasis using antibodies ELAN A and HBM 45. 

All patients were followed for at least 5 years after surgery, and distant metastasis and death events were recorded. The occurrence of distant metastasis was designated M_1_ and the absence M_0._ Distant metastasis-free survival (DMFS) was defined as the time from diagnosis to the onset of distant metastasis. Melanoma specific survival (MSS) was defined as the time from diagnosis to death from melanoma. Overall survival was calculated as the time from diagnosis to date of death related to melanoma or the last follow-up. The mean follow-up of the patents was 84 months (range, 20–168).

Written informed consent of all patients was obtained according to the Declaration of Helsinki.

### 4.2. Immunohistochemistry

Five µm serial paraffin-embedded sections were processed by immunohistochemistry. They were deparaffinized using xylene, rehydrated through graded ethanol, and treated with a microwave for heat-induced epitope retrieval in 10 mmol/L sodium citrate buffer (pH 6.0). The tissue sections were subsequently incubated with the monoclonal mouse antibody anti-CD9 (motility-related protein 1, NCL-CD9, Novocastra Laboratories, Newcastle, UK) overnight in humidified atmosphere at 4 °C.

The reaction was revealed using the streptavidin-biotin-peroxidase technique (EnVision Plus/HRP peroxidase kit; Dako Cytomation, Milan, Italy) according to Simonetti et al. [55]. CD9 staining was revealed by incubation with 3.3 diaminobenzidine (0.05 diaminobenzidine DAB in 0.05 mol/L Tris buffer, pH 7.6, and 0.01% hydrogen peroxide) (Sigma-Aldrich, Milano, Italy), a chromogen showing a brown precipitate at the location of the antibody.

For staining two antigens on a tissue-section using immuno-enzymatic detection double immunohistochemistry, two sets of antibodies were used: first antibody anti-CD9 and then anti-CD34 (clone HPCA1, Beckton Dickinson, Erembodegem, Belgium) for labelling blood vessels, or anti-D240 (prediluted; Signet Laboratories Inc, Dedham, MA) for highlighting lymphatic vessels. The sections designed for sequential double staining were previously stained for CD9, according to the procedure described above, and, upon completion of the first reaction, the antibody anti-CD34 or anti-D240 was applied overnight at 4 °C. Successively, the slices were immunostained by Envision peroxidase kit (Dako Cytomation, Milan, Italy) according to the standard procedure [43]. The sections were stained with DBA/Cobalt tablets (Sigma Fast DAB with Metal Enhancer, D0426, Sigma Aldrich, St. Luis, MO, USA) that yield an intense dark blue to bluish back color, easily distinguishable from brown DAB. In contrast to mono staining, which always ended with precipitated brown indicating CD9 expression, double staining produced CD34 or D240 stained dark blue and CD9 stained brown.

All the sections were counterstained with Mayer’s hematoxylin (Bio-Optica, Milan, Italy) and cover-slipped with Eukitt mounting medium (Electron Microscopy Sciences, Hatfield, PA, USA).

For all sections immunostained with specific antibodies, parallel sections were processed with isotype-matched control antibodies to confirm specificity.

Immunohistochemical staining of the markers was examined under a Nikon Eclipse E600 light microscope (Nikon, Düsseldorf, Germany) in at least 10 fields/samples at 400× magnification independently by two investigators (GG and GL) who had no previous knowledge of the histopathological classification. Agreement between the observers was always >95%.

The positivity of CD9-specific staining was assigned when the positive cells were >30% (quantified as a percentage of the total counted cells). In melanomas, the positivity was assigned according to the presence of CD9 positive tumor melanocytic clusters in certain regions of the sections. These tumor cell clusters were detected at a density of three to five in at least three microscopic fields, but they were never observed in tumors classified as negative.

### 4.3. Immunofluorescence

To validate the presence of CD9 in tissue from patients with melanoma, CD9 localization was also investigated by immunofluorescence. Paraffin-embedded sections were deparaffinized, hydrated with xylene and a graded alcohol series, and subjected to antigen retrieval with 10 mmol/L sodium citrate buffer (pH 6.0) in microwave. To reduce autofluorescence, samples were incubated with 0.1% Sudan Black B (Sigma-Aldrich, St Louis. USA) in 70% ethanol for 30 min then washed with PBS 1X with 0.03% Tween20. Sections were incubated with mouse antibody anti-CD9 (NCL-CD9, Novocastra Laboratories, Newcastle, UK) overnight in a wet box at 4 °C in the dark. Then, the anti-mouse fluorescein isothiocyanate (FITC) secondary antibody (Goxmo Fitch High Xads; Invitrogen, Carlsbad, CA, USA) was added and incubated at room temperature for 1 h. Slides were then incubated with DAPI (Invitrogen, Carlsbad, CA, USA) for 10 min for nuclear staining, washed and mounted onto glass slides using Vectashield mounting medium (Vector Laboratories, Inc., Burlingame, CA, USA). Negative controls were performed by omitting the primary antibody. Stained sections were examined and photographed using a fluorescence microscope (Nikon Eclipse E600, Nikon Instruments Europe B.V., Badhoevedorp, The Nederlands).

For staining two antigens on a tissue-section, double immunofluorescence was performed using two sets of antibodies: first antibody anti-CD9 and then anti-CD34 (clone HPCA1, Beckton Dickinson, Erembodegem, Belgium) for labelling blood vessels, or anti-D240 (prediluted; Signet Laboratories Inc, Dedham, MA, USA) for highlighting lymphatic vessels. The sections were stained for CD9, according to the immunofluorescence procedure described above, and, upon completion of the first reaction, the antibody anti-CD34 or anti-D240 was applied overnight at 4 °C. Successively, anti-mouse rhodamine-conjugated secondary antibody (Goxmo, Tritch Affinity, Invitrogen, Carlsbad, CA, USA) was added at room temperature for 1 h. The vessels positive to CD9 (green labelled) and anti-CD34 or anti-D240 (red labelled) showed colocalization of these markers at the same site and the positive vessels appeared stained orange. Slides were then incubated with DAPI (Invitrogen, Carlsbad, CA, USA).

### 4.4. Statistical Analysis

Statistical analyses were carried using the SPSS 16 package (SPSS Inc., Chicago, IL, USA) and GraphPad Prism 9 (GraphPad Software, San Diego, CA, USA). Differences between mean values in different categories were analyzed by paired Student t test, unpaired Student t test analysis of variance (ANOVA) test followed by Bonferroni test. Chi-square test was performed for the basic clinical-pathological characteristics of the patient cohorts. The Kaplan–Meier method was used to describe differences in survival for the different expression groups; statistical analysis was carried out by use of a log-rank test. *p* < 0.05 was considered to indicate a statistically significant difference.

A logistic regression model was used for analyzing risk factors for SNB positivity. The results of the logistic regression model were presented as odds ratio (OR), 95% confidence interval (CI) and *p*-value. All variables were evaluated by multivariate Cox regression to identify independent predictors of MSS. Insignificant prognostic factors were excluded from the model through Wald-backward elimination. Probability values of <0.05 were considered as significant.

## Figures and Tables

**Figure 1 ijms-23-04775-f001:**
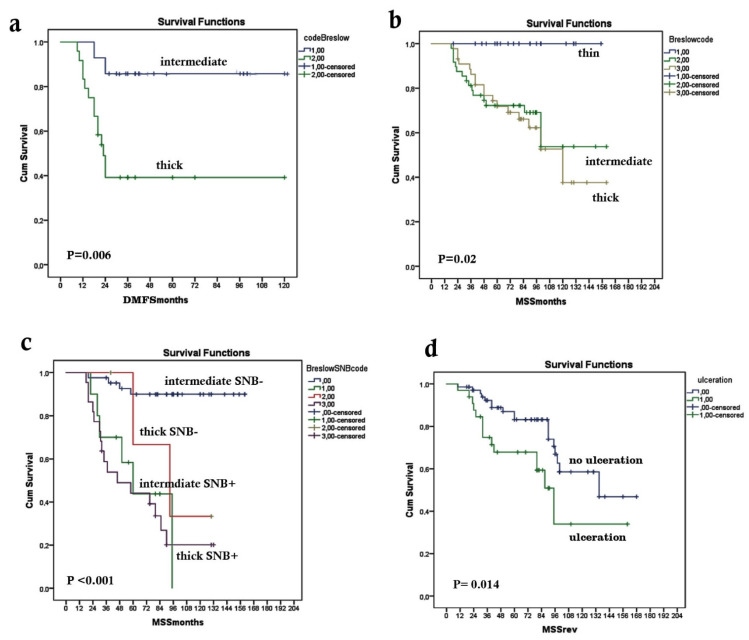
Kaplan–Meier analysis of Distant Metastasis Free Survival (DMFS) and Melanoma Specific Survival (MSS). (**a**) Patients with thick melanomas were significantly associated with shorter DMFS respect to the patients with intermediate melanoma. (**b**) Breslow thickness was significantly associated with MSS. (**c**) Breslow thickness >1 mm (intermediate and thick melanoma) associated with positive sentinel node biopsy (SNB) and (**d**) tumor ulceration presence were significant negative prognostic factors for MSS.

**Figure 2 ijms-23-04775-f002:**
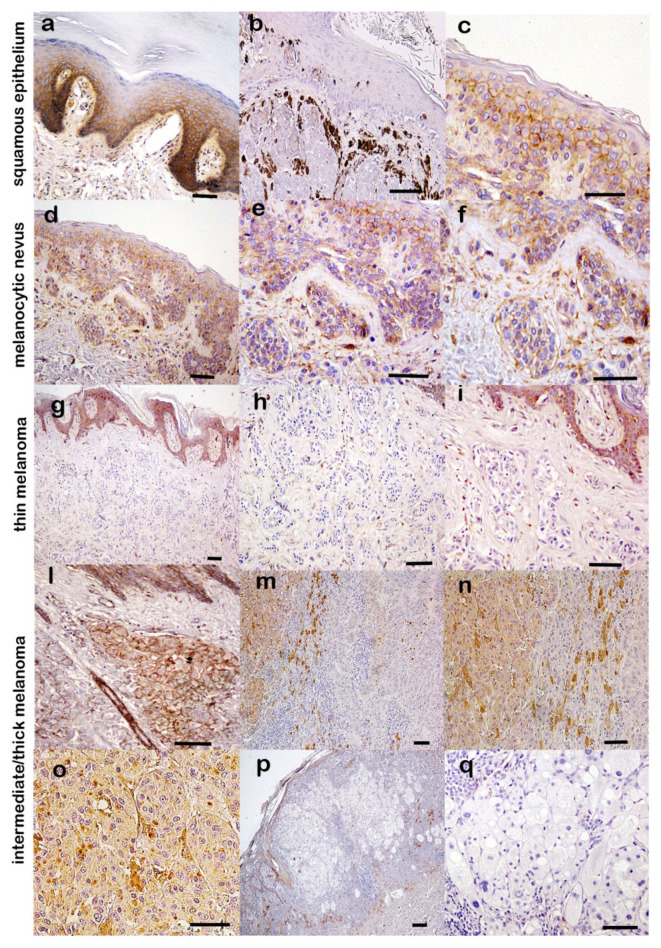
CD9 expression in melanocytic nevi and melanomas. (**a**–**c**) CD9 staining in the squamous epithelium, (**b**) CD9 staining was often not conserved in the epithelium from which the tumors arose. (**d**–**f**) Homogeneous CD9 staining in melanocytic nevus. (**g**–**i**) Loss of CD9 expression in thin melanoma. (**l**–**o**) CD9 staining was often re-expressed in small clusters of tumor melanocytic cells in intermediate and thick melanomas. (**p**,**q**) Thick melanoma negative for CD9. (immuno-peroxidase, scale bars 50 µ).

**Figure 3 ijms-23-04775-f003:**
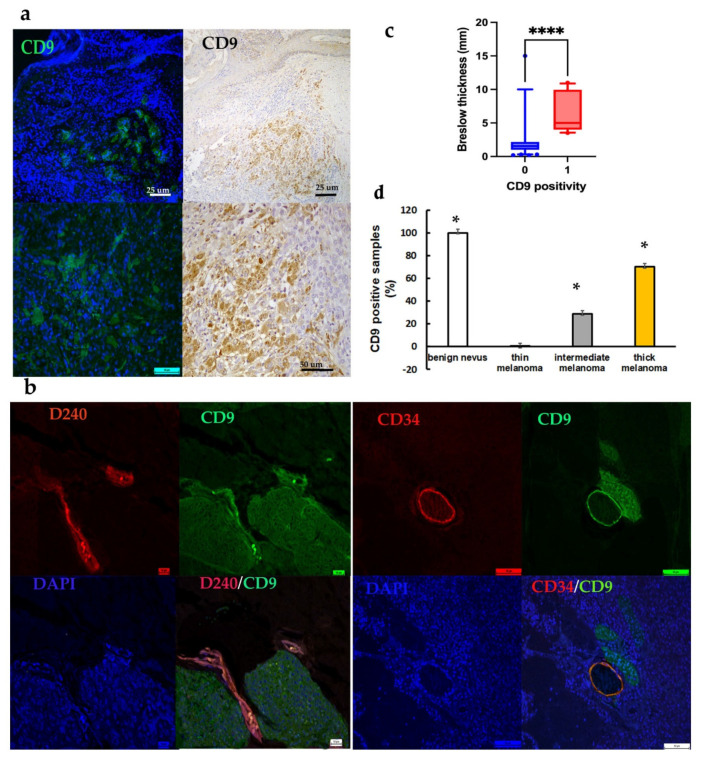
(**a**) Similar staining pattern of CD9 in positive tumor clusters using immunofluorescence and immunohistochemistry. (**b**) Endothelial vessels (CD34+) and lymphatic vessels (D240+) resulted CD9 positive using double immunofluorescence (orange staining). (**c**) CD9 expression related to Breslow thickness in cutaneous melanoma (0: negativity; 1: positivity; Student t test, **** *p* < 0.0001). (**d**) Statistical representation of CD9 positivity in benign nevi and cutaneous melanomas grouped according to Breslow thickness (ANOVA and Bonferroni test, * *p* < 0.05).

**Figure 4 ijms-23-04775-f004:**
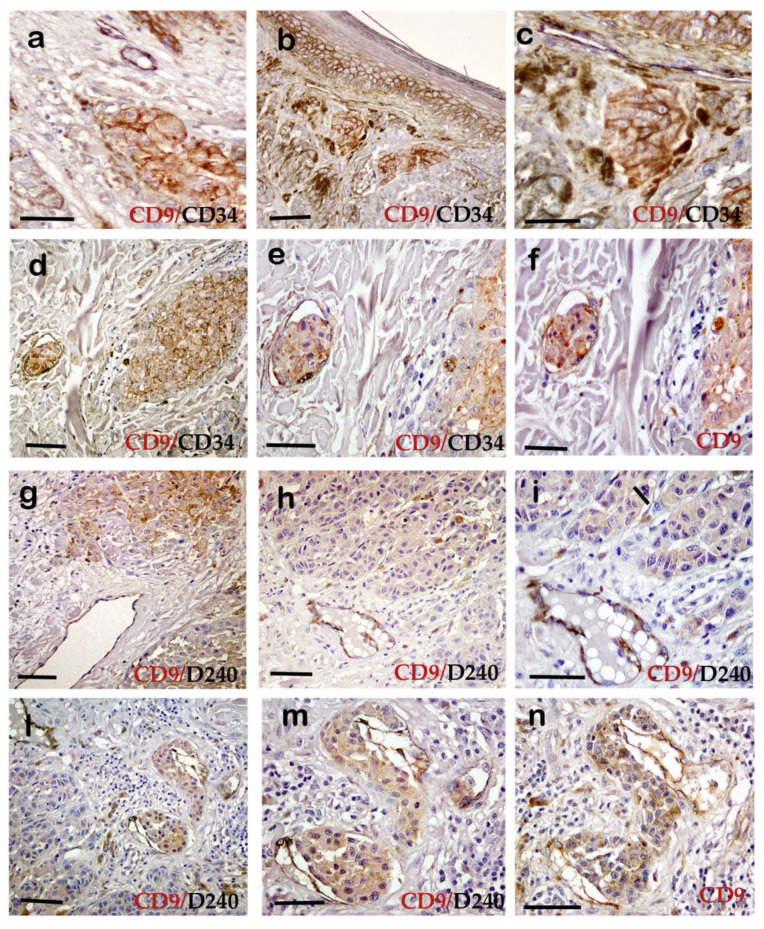
CD9 re-expression in positive tumor clusters near or inside blood and lymphatic vessels by immunohistochemical double staining. (**a**–**f**) Tumor melanocytic cells and blood vessels stained by CD9/CD34 double staining. (**g**–**i**,**l**–**n**) Tumor melanocytic cells and lymphatic vessels stained by CD9/D240 double staining. (immuno-peroxidase scale bars 50 µ).

**Figure 5 ijms-23-04775-f005:**
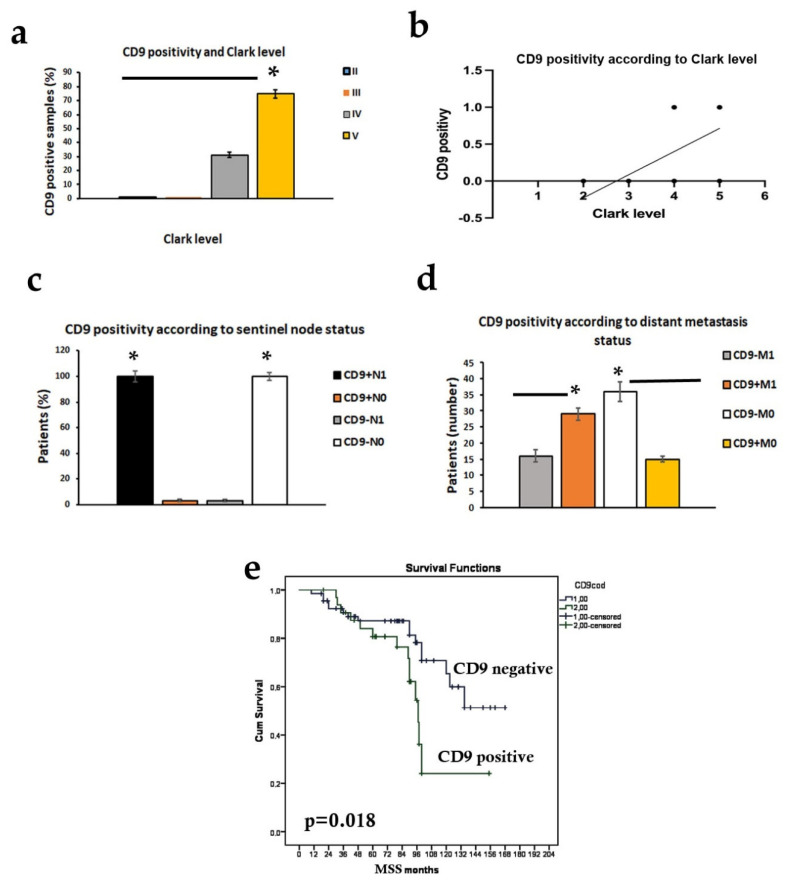
(**a**) Statistical representation of CD9 expression according to Clark level. CD9 staining was lost in II and III Clark levels and re-expressed in IV and V levels (* *p* < 0.05). (**b**) Correlation between CD9 and Clark level (*p* < 0.001; r = 0.753, Spearman rank correlation). (**c**,**d**) Statistical representation of CD9 expression according to sentinel node and distant metastasis status (* *p* < 0.001); CD9+, positive to CD9; CD9-, negative to CD9; N0, sentinel lymph node negative; N1, sentinel lymph node positive; M0, absence of distant metastasis; M1, presence of distant metastasis. (**e**) Kaplan–Meier survival curves showed a significant association between CD9 expression and melanoma specific survival in patients with melanoma (*p* = 0.018).

**Table 1 ijms-23-04775-t001:** Patient characteristics.

Characteristic	Patient Number *p*-Value
*Gender*	
Male	75
Female	65
*Age*	
<40 years	50
≥40 years	90
*Melanocytic nevus*	20
*Primary melanoma*	120
*Breslow thickness*	
≤1.0 mm (thin)	24
1.1–4.0 mm (intermediate)	56
>4.0 mm (thick)	40
*Clark level*	
II	8
III	16
IV	64
V	32
*Tumor ulceration presence*	
Thin melanoma	-
Intermediate melanoma	12 (21.4%)
Thick melanoma	22 (55%) <0.01 *
*Sentinel lymph node biopsy (SNB) performed*	96
Positive sentinel nodes	44
Intermediate melanoma	16 (29%)
Thick melanoma	28 (70%) <0.001 *
*Distant metastasis (M_1)_*	
Thin melanoma M_1_	-
Intermediate melanoma M_1_	20 (35%)
Thick melanoma M_1_	25 (62%)

* *p*-value by Chi square test.

**Table 2 ijms-23-04775-t002:** Predictive values of high-risk pathological and clinical features for positive sentinel lymph node biopsy (SNB) in patients with intermediate and thick melanomas.

SNB Risk Factor	No. Cases ^a^	No. Controls ^b^	Odds Ratio	95% CI
*Gender*				
Female	20	20	1	Referent
Male	24	32	0.75	0.332–1.694
*p* = 0.49				
*Age*				
<40	15	15	1	Referent
≥40	29	37	0.783	0.330–1.862
*p* = 0.58				
*Breslow thickness*				
Intermediate	16	40	1	Referent
Thick	28	12	5.833	2.394–14.216
***p* < 0.001**				
*Tumor ulceration*				
Absence	12	50	1	Referent
Presence	22	12	7.639	2.971–19.639
***p* < 0.0001**				
*CD9 expression*				
Negativity	0	52	1	Referent
Positivity	44	0	8505	1651.8–437,916.42
***p* < 0.0001**				

^a^ Patients with SNB positive; ^b^ Patients with SNB negative; Abbreviation: SNB, sentinel lymph node biopsy; CI, confidence interval. The *p*-values that are statistically significant are highlighted in bold.

**Table 3 ijms-23-04775-t003:** Multivariate Cox regression associated with melanoma specific survival (MSS).

Variable	HR	95% CI (HR)	*p*-Value
*Gender*			
Female	1	Referent	
Male	0.965	1.019–1.722	0.864
*Age*			
<40	1	Referent	
≥40	0.992	334–2.468	0.561
*Breslow thickness*			
Thin	1	Referent	
Intermediate	1.15	0.321–4.120	0.761
Thick	1.65	0.641–4.239	0.298
*Tumor ulceration*			
Absence	1	Referent	
Presence	3.741	1.377–10.164	**0.010**
*SNB status*			
Negative	1	Referent	
Positive	17.323	4.229–70.958	**0.0001**
Not recommended ^a^	1.105	0.448–2.725	0.658
*CD9 expression*			
Negative	1	Referent	
Positive	13.077	2.896–59.046	**0.0001**

Note: Regression model with stepwise Wald-backward adjusted for gender, age, Breslow thickness, tumor ulceration, SNB status, and CD9 expression. Abbreviations: CI, confidence interval; HR, hazard ratio; SNB, sentinel lymph node biopsy. ^a^ Patients with thin melanoma: SNB is not recommended; The *p*-values that are statistically significant are highlighted in bold.

**Table 4 ijms-23-04775-t004:** Expression of CD9 in melanocytic nevi and primary melanomas.

Patient’s Characteristics (No. Patients)	CD9+ No. Patients (%)	CD9- No. Patients (%)
*Histology*		
Melanocytic nevus (20)	20 (100%)	-
Primary melanoma (120):		
*Breslow thickness*		
Thin melanoma (24)	-	24 (100%)
Intermediate melanoma (56)	16 (29%)	40 (71%)
Thick melanoma (40)	28 (70%)	12 (30%)
*Clark level*		
II (8)	-	8 (100%)
III (16)	-	16 (100%)
IV (64)	20 (31%)	40 (69%)
V (32)	24 (75%)	8 (25%)
*Sentinel lymph node biopsy status*		
Positive (44)	44 (100%)	-
Negative (52)	-	52 (100%)
*Distant metastasis (M) **		
Presence M_1_ (45)	29 (64%) **	16 (36%) **
Absence M_0_ (51)	15 (29%) **	36 (71%) **

* Distant metastasis in intermediate and thick melanomas; ** A chi-square test of independence was performed to examine the relation between CD9 positivity and M status. The relation between these variables was significant, χ^2^ = 10.5, *p* < 0.001.

## Data Availability

The data presented in this study are available in this article.

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
