# Peer review of "Tetraspanin CD9 Expression Predicts Sentinel Node Status in Patients with Cutaneous Melanoma"

_ijms, 2022, doi:10.3390/ijms23094775_

Round 1

Reviewer 1 Report

In this manuscript the authors describe the immunohistochemistry analyses of the tetraspanin CD9 in melanocytic nevi and primary cutaneous melanomas. The data show homogeneous staining of CD9 in nevi, complete loss in thin melanomas and expression in 46% of intermediate and thick melanomas, particularly in small tumor clusters located invasion sites near or inside the blood or lymphatic vessels. Additionally, they report a direct association between CD9 expression and the presence of distant metastasis suggesting specific function of CD9 in vascular dissemination during late tumor progression. The data are interesting although the function of CD9 was not explored.

Reviewer 2 Report

Dear editor,

it was with interest that I read the manuscript by Guendalina et al concerning the prognostic role of CD9 in melanoma. Please see my comments below:

  1. CD9 is one of the hallmark markers for extracellular vesicles (CD9, C63 and CD81), and the role of EVs in creating pre-metastatic niches have been shown by Peinado et al in two Nature papers, there is also other papers discussing EVs in melanoma, and this could definitely be brought to attention in the introduction as a mechanism why CD9 could be a prognostic marker. This is supported also by the finding that CD9 was found outside of the tumour…
  2. It is stated that the sentinel node was identified in 96 patients, but in how many patients was it performed? I guess that the indication was Breslow >1mm?
  3. The analysis showing that SLN positivity was more common in intermediate and thick vs thin is most probably explained by the fact that SLN was not performed in thin melanomas? Or am I misunderstanding the analysis?
  4. Survival should be reported as median survival time, and not just percentages, alternatively that e.g. 5—year survival survival is given. This basically since the follow-up time most probably are different between all patients. The correct analysis to compare survival time is then a Kaplan-Meier analysis
  5. A p-value can not be 0.000 (all over the manuscript).
  6. Figure 2c and 4b. This does not seem to include all 120 patients? Since the x-value only can be 0 or 1, a linear regression is not the correct analysis, this is a dichotomous outcome. You could also revert the axis.
  7. Figure 4e. Here you report a Kaplan-Meier. The x-axis should be in 12 months interval. It does not seem to include all patients? There are quite few censored patients, and I only count to 5 deaths…. There should be 40.

Author Response

"Please see the attachment

Round 2

Reviewer 2 Report

Dear editor,

This is an extremely interesting paper, with very novel data! It should definitely be published, but the authors need to update the paper, preferably using a clinically oriented colleague and/or a statistician. Currently the manuscript has been somewhat improved, but honestly and sadly still badly written. The data that CD9 perfectly correlates to SNB status is almost too good to be true, and if it holds in independent validation, it would be a major break-through. I would even recommend that the authors would seek patent on the finding! There are several companies (SkyLine and Castle Biosciences) currently developing exactly these kind of analysis, and they are far from these data. So my comments below are to be seen as very friendly and encouraging, I am so eager of the results that I would have liked to write the paper myself. It will definitely be a highly cited paper in the future (if the data can be validated), I would even consider rewriting the title to “Tetraspanin CD9 expression predict sentinel node status in patients with cutaneous melanoma” and focus the paper of this extreme finding. A would also dreamed to see a logistic multivariable analysis including gender, age, Breslow, ulceration and CD9 status to see if CD9 was an independent factor for SNB status… and I would also add a Cox-regression with gender, age, Breslow, ulceration, SNB status and CD9 status for melanoma-specific survival (MSS). It would REALLY enhance the paper.

  1. Table 1. ”Sentinel lymphadenectomy” change to ”Sentinel lymph node biopsy performed”.
  2. Results, 2nd paragraph: Change ”The sentinel lymph node was identified in 96 patients with intermediate and thick melanoma.” to ”In the 96 patients with intermediate and thick melanoma, sentinel lymph node biopsy was performed”
  3. Change ”cases” to ”patients”
  4. In the Methods, it is stated that all specimens between 2005-2015 were retrieved, but it is nowhere stated how and why these specific samples were decided to be included in the study? I dont think that exactly 140 patients are all the patients during 10 years at this clinic? E.g. it is an unusually high percentage of patients with a positive sentinel node? Are there any selection bias?
  5. Table 1. It is stated that Students t-test was used to examine the difference between thin intermediate and thick melanomas on SN status… these are categorical variables and chi-square or similar should be used. Students can be used if data would be presented as mean Breslow thickness for e.g. positive vs negative, but that is not reported. The same goes for the comparisons of metastatic status, you can not compare categorical variables with Students t-test.
  6. Results: When reporting survival and recurrences, this has to be done using kaplan-Meier methodology taking censoring into account. Sentences should be like ”In the 120 patients with invasive melanoma, X (X%) developed distant metastasis after a median of Y months (95% CI X-X months). Of the 56 patients with intermediate melanoma, 20 patients developed distant metastasis after a median time of X months (95% CI X-X months), compared to Y months (95% CI X-X months) for the 25 of the 40 patients (62%) with thick melanomas (log-rank p=x.xx). Look at clinical papers reporting on clinical outcomes and rewrite this whole part of the Results. Also use a clinical statistician used to work with this kind of reporting.
  7. The sentence ”Overall survival did not differ in relation to melanoma thickness.” says absolutely nothing at all… what was the numbers??? Number of dead patients are not the correct way reporting these data, but rather survival time (incl censoring). You have to report like my example above.
  8. Table 2. Same problem with Students… wrong statistics. And if you want to compare distant metastasis… do this by a Kaplan-Meier analysis taking time to event and consoring into account.
  9. Figure 2c. It seems very strange to report CD9 positivity (dichotomous variable) and a continous variable (Breslow thichness) as a line diagram with a correlation test. I recommend that you present it with two boxplots for CD9+/- and Breslow on the y-axis. Then you can compare with Students t-test (if parametric).
  10. Figure 4e. Are all patients (including melanocytic nevi) included? Has to be described? That could explain why there is no difference since CD9 seemed to be present in those? Since CD9-status correlated EXACTLY to lymph node status, there should be a more marked survival difference? You could also analyse intermediate and thick melanomas in the same graph (4 lines thich SNB+/-, intermediate SNB+/-). What statistical test was used? If I would present this, I would not present only overall survival, bur rather melanoma-specific survival, since it is very unlikely that CD9 status would actually affect overall survival, but rather MSS. The x-axis should be in either 12-month interval (or years), and that you can change in SPSS by double clicking on the diagram and then the axis… take help from a statistician otherwise. The axis should also be updated and the legend as well.
  11. Discussion: Read the papers from e.g. Skyline and Castle on test prediciting SLN status (there are several validation cohorts published, e.g. Meeves et al or Johansson et al) and comment on the role of CD9 to predict SLN status.

Author Response

Response to Reviewer 2 Comments

We are grateful to the reviewer for the comments, which we feel have considerably improved our manuscript. We highlighted the changes in the text in yellow.

CD9 perfectly correlates to SNB status is almost too good to betrue, and if it holds in independent validation, it would be a major break-through. I would even recommend that the authors would seek patent on the finding! There are several companies (SkyLine and Castle Biosciences) currently developing exactly these kind of analysis, and they are far from these data. So my comments below are to be seen as very friendly andencouraging, I am so eager of the results that I would have liked to write the paper myself. It will definitely be a highly cited paperin the future (if the data can be validated), I would even consider rewriting the title to “Tetraspanin CD9 expression predict sentinelnode status in patients with cutaneous melanoma” and focus the paper of this extreme finding. A would also dreamed to see a logistic multivariable analysis including gender, age, Breslow,ulceration and CD9 status to see if CD9 was an independent factor for SNB status… and I would also add a Cox-regression with gender, age, Breslow, ulceration, SNB status and CD9status for melanoma-specific survival (MSS). It would REALLY enhance the paper.

Response  According to the reviewer suggestion, we changed the manuscript title. In addition, we performed logistic multivariate and Cox -regression analysis and the results are shown in the Tables 2 and 3 (new tables).

Point 1.  Table 1. ”Sentinel lymphadenectomy” change to ”Sentinel lymph node biopsy performed”.

Response to point 1. We corrected according to the reviewer suggestion

Point.2   Results, 2nd paragraph: Change ”The sentinel lymph node was identified in 96 patients with intermediate and thick melanoma.” to ”In the 96 patients with intermediate and thick melanoma, sentinel lymph node biopsy was performed”

Response to point 2.  We corrected according to the reviewer suggestion

Point 3.  Change ”cases” to ”patients”

Response to point 3. We corrected according to the reviewer suggestion

Point 4.   In the Methods, it is stated that all specimens between 2005-2015 were retrieved, but it is now here stated how and why these specific samples were decided to be included in the study? I dont think that exactly 140 patients are all the patients during 10 years at this clinic? E.g. it is an unusually high percentage of patients with a positive sentinel node?Are there any selection bias?

Response to point 4. We added the inclusion/ exclusion criteria for the selection of melanoma patients in Materials and Methods, 4.1. Sample collection section.

Point 5. Table 1. It is stated that Students t-test was used to examine the difference between thin intermediate and thick melanomas on SN status… these are categorical variables and chi-square or similar should be used. Students can be used if data would be presented as mean Breslow thickness for e.g. positive vs negative, but that is not reported. The same goes for the comparisons of metastatic status, you cannot compare categorical variables with Students t-test.

Response to point 5.  We revised the statistical analysis.  We performed chi-square test and revised Table 1.

Point.6. Results: When reporting survival and recurrences, this has to be done using kaplan-Meier methodology taking censoring into account. Sentences should be like ”In the 120 patients with invasive melanoma, X (X%) developed distant metastasis after a median of Y months (95% CI X-X months).Of the 56 patients with intermediate melanoma, 20 patients developed distant metastasis after a median time of X months (95% CI X-X months), compared to Y months (95% CI X-X months) for the 25 of the 40 patients (62%) with thick melanomas (log-rank p=x.xx). Look at clinical papers reporting on clinical outcomes and rewrite this whole part of the Results. Also use a clinical statistician used to work with his kind of reporting.

Response to point 6. We performed Kaplan-Meier analysis and revised Results, Patient characteristics section according to the reviewer suggestions. We showed the Kaplan-Meier curves in a new Figure 1.

Point 7. The sentence ”Overall survival did not differ in relation to melanoma thickness.” says absolutely nothing at all… wha twas the numbers??? Number of dead patients are not the correct way reporting these data, but rather survival time (incl censoring). You have to report like my example above.

Response to point 7. According to the reviewer suggestion, we reported melanoma specific survival time (MSS) for each patient group by Kaplan-Meier methodology (Results, Patient characteristics section). We showed Kaplan curves for MSS in Figure 1.

Point 8. Table 2. Same problem with Students… wrong statistics. And if you want to compare distant metastasis… do this by a Kaplan-Meier analysis taking time to event and censoring in to account.

Response to point 8. We revised the statistical analysis.  We performed chi-square test and revised Table 2 (now Table 4). We performed Kaplan-Meier analysis of Distant Metastasis Free Survival (DMFS) shown in Figure 1 and we rewrote the paragraph on distant metastasis (Results, Patient characteristics section) according to the reviewer suggestions.

Point.9  Figure 2c. It seems very strange to report CD9 positivity (dichotomous variable) and a continous variable (Breslow thichness) as a line diagram with a correlation test. I recommend that you present it with two boxplots for CD9+/-and Breslow on the y-axis. Then you can compare with Students t-test (if parametric).

Response to point 10. We revised statistical analysis according to the reviewer suggestion. We revised the Figure 2c (now Figure 3c).

Point 10. Figure 4e. Are all patients (including melanocytic nevi) included? Has to be described? That could explain why there is no difference since CD9 seemed to be present in those?Since CD9-status correlated EXACTLY to lymph node status,there should be a more marked survival difference? You could also analyse intermediate and thick melanomas in the same graph (4 lines thich SNB+/-, intermediate SNB+/-).What statistical test was used? If I would present this, I would not present only overall survival, bur rather melanoma-specific survival, since it is very unlikely that CD9 status would actually affect overall survival, but rather MSS. The x-axis should be in either 12-month interval (or years), and that you can change in SPSS by double clicking on the diagram and then the axis… take help from a statistician otherwise.The axis should also be updated and the legend as well.

Response to point 10. We have specified in the legend that the Figure 4e (now 5e) only concerns MSS of the patients with melanoma. According to the reviewer suggestion we performed Kaplan -Meier analysis for MSS instead of OS (Figure 5e), obtaining a significant association between CD9 expression and patient survival. In addition, we analyzed MSS according to the Breslow depth and SNB status in the same graph (4 lines thicK SNB+/-, intermediate SNB+/-) by Kaplan methods (Figure 1 c) and we reported the results in Patient characteristics section.

Point 11. Discussion: Read the papers from e.g. Skyline and Castle on test prediciting SLN status (there are several validation cohorts published, e.g. Meeves et al or Johansson et al) and comment on the role of CD9 to predict SLN status.

Response to point 11. By following the reviewer suggestion, we revised the discussion.

Round 3

Reviewer 2 Report

Thanks for the revised manuscript which has been improved dramatically. However, there are still things to be fixed.

  1. You changed the legend in Figure 5e. But it still says "Kaplan- Meier survival curves showed no relation between CD9 expression and overall survival (P=0.192)"... but it is MSS not OS, the p-value is 0.018 and not 0.192, and there is a difference?? Three errors in one line.
  2. Table 2. Have you used Cox-regression to identify factors for a positive SNB? Cox-regression is a "time-to-event" analysis and to use it to identify a binary outcome (pos/neg) is not just wrong, it means that you have absolutely no clue what you are doing. You have to to a logistic regression, reporting odds ratios, and not hazard ratios. Another major point is that in multivariate analysis, you must tell what is the reference. In e.g. Table2, what does a HR of 0.52 for Gender meaning. Is it women or men that have a lower risk? And a HR of 0.44 for CD9 would mean that CD9 positivity would LOWER the risk of a positive SNB with 66%... and that is not coherent with your other results.
  3. In Table 3 you have to report reference categories for all non-continues variables. And e.g. that ulceration would lower the risk of death seems very unlikely (you have probably switched the reference.
  4. I honestly believe that you need a proper statistical consultation that really verifies all of the statistics and how it is reported. There is almost no statistical analysis performed in this paper that has been correct, even though it has improved after my comments. But the work of a reviewer is not to go through all details and do a revision, that has to be done and vetted for by the authors according to strict scientific stringency. You can not just take a test and run it through SPSS not knowing what you are doing, that is sadly the true impression I get. 

Author Response

Response to Reviewer 2 Comments

We are grateful to the reviewer for the comments which have certainly improved our experimental study by expanding some clinical aspects. We reviewed the statistical analysis with a clinical statistician. We highlighted the changes in the text in yellow.

Point 1. You changed the legend in Figure 5e. But it still says "Kaplan- Meier survival curves showed no relation between CD9 expression and overall survival (P=0.192)"... but it is MSS not OS, the p-value is 0.018 and not 0.192, and there is a difference?? Three errors in one line.

Response point 1.  We changed the legend in Figure 5e. Unfortunately, this is a carelessness. In the revised version we have corrected the result but we have not corrected the legend which has remained the same as in the previous version.

Point 2: Table 2. Have you used Cox-regression to identify factors for a positive SNB? Cox-regression is a "time-to-event" analysis and to use it to identify a binary outcome (pos/neg) is not just wrong, it means that you have absolutely no clue what you are doing. You have to a logistic regression, reporting odds ratios, and not hazard ratios. Another major point is that in multivariate analysis, you must tell what is the reference. In e.g. Table2, what does a HR of 0.52 for Gender meaning. Is it women or men that have a lower risk? And a HR of 0.44 for CD9 would mean that CD9 positivity would LOWER the risk of a positive SNB with 66%... and that is not coherent with your other results.

Response to point 2. We reviewed the statistical analysis with a clinical statistician. We performed a logistic regression analysis shown in Table 2.

Point 3. In Table 3 you have to report reference categories for all non-continues variables. And e.g. that ulceration would lower the risk of death seems very unlikely (you have probably switched the reference.

Response to point 3.  We revised Cox regression analysis. We revised the Table 3. 

Point 4.   I honestly believe that you need a proper statistical consultation that really verifies all of the statistics and how it is reported. There is almost no statistical analysis performed in this paper that has been correct, even though it has improved after my comments. But the work of a reviewer is not to go through all details and do a revision, that has to be done and vetted for by the authors according to strict scientific stringency. You can not just take a test and run it through SPSS not knowing what you are doing, that is sadly the true impression I get. 

Response to point 4.  We reviewed the whole statistical analysis with the help of a clinical statistician. We added the Kaplan Meier analysis to study the association between tumor ulceration and MSS. We added the results in the Patient Characteristics section and Figure 1d.

We hope that our manuscript is now suitable for publication, considering that our study was primarily an experimental study. We believe that other studies are needed to confirm our findings, especially our clinical results, with the involvement of a larger number of melanoma patients.

Round 4

Reviewer 2 Report

I now think that the major revisions needed are performed, and I can jus congratulate the authors for this very interesting paper!